# Heart Organoids and Engineered Heart Tissues: Novel Tools for Modeling Human Cardiac Biology and Disease

**DOI:** 10.3390/biom11091277

**Published:** 2021-08-26

**Authors:** Yonatan R. Lewis-Israeli, Aaron H. Wasserman, Aitor Aguirre

**Affiliations:** 1Division of Developmental and Stem Cell Biology, Institute for Quantitative Health Science and Engineering, Michigan State University, East Lansing, MI 48823, USA; israeli1@msu.edu (Y.R.L.-I.); awasserm@msu.edu (A.H.W.); 2Department of Biomedical Engineering, College of Engineering, Michigan State University, East Lansing, MI 48823, USA

**Keywords:** heart, organoid, pluripotent stem cell, cardiovascular, stem cell, development, cardiovascular disease, self-organization, directed assembly

## Abstract

Organoids are three-dimensional in vitro cell constructs that recapitulate organ properties and structure to a significant extent. They constitute particularly useful models to study unapproachable states in humans, such as embryonic and fetal development, or early disease progression in adults. In recent years organoids have been implemented to model a wide range of different organs and disease conditions. However, the technology for their fabrication and application to cardiovascular studies has been lagging significantly when compared to other organoid types (e.g., brain, pancreas, kidney, intestine). This is a surprising fact since cardiovascular disease (CVD) and congenital heart disease (CHD) constitute the leading cause of mortality and morbidity in the developed world, and the most common birth defect in humans, respectively, and collectively constitute one of the largest unmet medical needs in the modern world. There is a critical need to establish in vitro models of the human heart that faithfully recapitulate its biology and function, thus enabling basic and translational studies to develop new therapeutics. Generating heart organoids that truly resemble the heart has proven difficult due to its complexity, but significant progress has been made recently to overcome this obstacle. In this review, we will discuss progress in novel heart organoid generation methods, the advantages and disadvantages of each approach, and their translational applications for advancing cardiovascular studies and the treatment of heart disorders.

## 1. Introduction

Cardiovascular disease (CVD) is the leading cause of mortality both in the United States and in the rest of the developed world [1]. While traditional animal models, such as mice and rats, have proven useful for studying CVD etiology and pathogenesis, there is a critical need for human-derived models to be able to fully translate these findings to the clinic. The in vitro generation of complex three-dimensional cardiac tissues (“miniature hearts”) that faithfully recapitulate human cardiac cell type composition, structure, and function will be paramount in advancing basic research studies as well as pre-clinical CVD research. In recent years, human pluripotent stem cells (hPSCs) and primary cells have been successfully used to create organ-like (organoid) models of the brain [2,3,4,5], lung [6,7,8], liver [9,10,11], pancreas [12,13,14], intestine [15,16,17], colon [18,19,20], and kidney [21,22,23]. However, there has been less progress on human heart organoids, likely due to its significant structural complexity and challenges associated with its biomechanics and vascularization. The potential applications of human heart organoids are wide-ranging. Because they recapitulate the structural complexity of the heart and follow developmental tissue maturation steps, they can be used to study human cardiac development in a dish and model congenital heart diseases. More mature cardiac organoids can be used to study heart disease in adults and the pathophysiology of myocardial infarction and cardiac injury. In preclinical studies, heart organoids might prove to be a valuable screening tool for determining the therapeutic or cardiotoxic effects of drugs. Patient-specific hPSCs can be used to generate organoids that can greatly advance the field of personalized medicine [24]. In the long term, heart organoids might become viable sources of transplant tissue. For all these applications, however, creating more sophisticated and reproducible human heart organoids is a key necessary preliminary step that still presents significant challenges.

There are two different general schemes for producing heart organoids, directed assembly and self-organization (also referred to as self-assembly, Figure 1). It is worth noting that throughout this review, we will be referring to directly assembled and self-organizing tissues as organoids indistinctly in the most general sense of the term, albeit some authors may prefer the specific distinction of engineered heart tissues (EHT) vs. organoids proper. Our rationale for this choice is the fact that both methods attempt to recreate constructs with tissue-like structure and functionality. In addition, even though the 3D tissues generated in some of the studies we will review may classify more as EHTs than organoids, we will use the nomenclature that the original authors used out of respect for their work.

In directed assembly, stem cell-derived differentiated cardiac cells or primary cardiac cells are co-cultured and seeded onto a scaffold or bioengineered device that allows them to develop their 3D structure. In some cases, this structure is obtained by simply growing the cells on an extracellular matrix (ECM) hydrogel or in microwell molds, but it can also be through more complex means, such as using a biocompatible elastic pillar or a custom-designed bioreactor [25,26]. The earliest reports attempting to create in vitro organ-like heart tissues (sometimes referred to throughout the literature as engineered heart tissues or EHTs) employed cells obtained from primary sources, such as rat and mouse tissues, and aggregated them into a spherical structure, producing what are commonly known as cardiac spheroids. Originally made from only one or two relevant cardiac cell types in co-culture, the cells in these constructs would stick together via cell–cell adhesions and organize into a loosely spherical mass [27,28,29,30,31,32,33,34,35]. Other early studies utilized micropatterning of cardiac cells to form non-spherical structures that were more elongated and linear in shape [36,37]. While the above pioneering constructs were utilized for disease modeling and engraftment with a certain degree of success, they did not develop noteworthy morphological or anatomical complexity and lacked many of the cell types found in the human heart. More recently, however, EHTs have experienced significant innovations raising their complexity and modeling capabilities [38,39,40,41,42], as discussed in more detail below.

In recent years, self-organization, a technique that is based on our growing understanding of heart embryonic development and advances in stem cell culture, has gained popularity. In this approach, stem or progenitor cells form the 3D structure autonomously, with minimal exogenous intervention, due to the presence of morphogens and growth factors in the media and ECM [43]. Driving developmental cues are provided in the medium in a carefully time-controlled stepwise fashion to induce differentiation into heart lineages. The established cardiac stem cell lineages produced release their own local morphogens and trigger higher levels of structure and organization quickly. Organoids generated via this method can develop very high levels of complexity, with most or all of the cell types found in the heart, and spontaneously acquire relevant rudimentary anatomical morphology, such as chamber organization and atrioventricular specification [44,45,46,47]. The cost of this approach is lower control over the final result in comparison to organoids fabricated via direct assembly.

## 2. Directed Assembly of Human Heart Organoids

Several studies have been published demonstrating the directed assembly of three-dimensional heart organoids in vitro. These methods differ widely in terms of differentiation protocols, cardiac cell type compositions, and potential applications. However, they all utilize some type of scaffold, biomaterial, or ECM component to drive the formation of 3D cell aggregates from dissociated cell suspensions. One example of a directed organoid differentiation protocol that has been studied extensively in recent years is the heart dynamometer (Heart-Dyno) microtissue platform [40]. At its core, this method involves taking pre-differentiated cardiac cells and allowing them to condense around two elastic scaffolds, forming a circular structure. The Heart-Dyno is a 96-well plate with a custom-designed culture insert inside each well containing two polydimethylsiloxane (PDMS) elastomeric posts spaced 1.0 mm apart. In the first study utilizing this system, Mills et al. took hiPSC-derived cardiomyocytes and stromal cells at a ~70:30 ratio, mixed them with collagen I and Matrigel (synthetic ECM), and added this mixture to the culture insert. They then allowed the cells to gel for ~30–45 min at 37 °C before immersing them in culture media, which led to them condensing around the two elastic poles within two days, forming human cardiac organoids (hCOs). The presence of the posts stabilizes hCO contraction and allows contraction force to be measured in response to specific inotropic agents [40].

The applications of the Heart-Dyno platform are numerous, as it provides a 96-well device for functional screening of agents that promote organoid growth, maturation, and function. Mills et al. [40] first set out to study cardiac maturation by supplementing the hCOs with palmitate, causing them to switch their metabolism from glycolysis to fatty acid oxidation. They found that fatty acids induced the organoids to display more mature contractile, electrophysiological, and transcriptional properties. In addition, the maturation medium caused cardiomyocytes (CMs) to proliferate less and activate the DNA damage response, which drives terminal CM differentiation [40]. In another study, the Heart-Dyno system was utilized to screen 105 small molecules for their pro-regenerative capacity in immature and mature hCOs [48]. The authors identified two compounds that not only increased CM proliferation but also had no detrimental effects on hCO contractile function. Transcriptomic analysis revealed that these hit molecules activated the cell cycle network and the mevalonate pathway, which synthesizes the precursors for cholesterol and steroid hormones [49], thereby suggesting a mechanism for inducing pro-regenerative effects in human hearts [48]. One final application of the Heart-Dyno is for the study of cardiac damage that occurs as a result of COVID-19 infection. In this recent study [50], the effects of candidate pro-inflammatory cytokines on hCO function were investigated, leading to the identification of a cocktail of compounds that induced diastolic dysfunction. Mills et al. found that bromodomain-containing protein 4 (BRD4) is activated in response to this “cytokine storm” and that bromodomain extra terminal inhibitors (BETi) improved hCO function. Importantly, one specific BETi (INCB054329) also prevented SARS-CoV-2 infection of cardiac cells, a finding that can have important public health implications in the near future [50].

A very similar hCO platform (with CMs differentiated from hESCs instead of hiPSCs) was also recently used to develop an in vitro cardiac injury model [51]. In this study, the authors induced cryoinjury in cardiac organoids by applying a dry ice probe directly to their surfaces, leading to localized cell death and reduced contractile function. Remarkably, the injured hCOs displayed high levels of CM proliferation and full functional recovery after two weeks, further suggesting that they can be used to study heart regeneration in a dish [51].

Recently, an important issue that directly-assembled organoids have been used to model is cardiac tissue maturation. In one landmark study, Ronaldson-Bouchard et al. [25] made 3D human cardiac tissues by growing cell populations of 75% hiPSC-derived CMs and 25% human dermal fibroblasts and encapsulating them in a fibrin ECM. They then dispensed these mixtures into a polycarbonate-based tissue bioreactor platform consisting of 12 tissue culture wells with flexible elastomeric pillars in each well. After the cardiac tissues formed around these pillars (approximately one week of culture), the authors subjected them to gradually increasing electrical stimulation, with the goal of forming structures that were as mature as possible. Indeed, after three weeks of intensity training, the human cardiac tissues displayed enhanced cardiac ultrastructure, oxidative metabolism, T-tubule formation, and calcium handling dynamics, thereby demonstrating the potential of this system for modeling adult cardiac disease [25]. Another platform that utilized electrical stimulation to mature three-dimensional heart tissues is the Biowire system [52,53]. Here, hESC-CMs and supporting cells were seeded into template PDMS channels containing a sterile surgical suture and embedded in type I collagen gel. Over the course of 7 days, the cardiac cells remodeled around the suture, forming 3D rectangular tissues with aligned CMs (biowires). Throughout this time, the biowires were matured via gradually increasing electrical stimulation at low frequency (1 Hz to 3 Hz) or high frequency (1 Hz to 6 Hz). After stimulation, the biowires showed increased CM myofibril organization, elevated action potential conduction velocity, and improved calcium handling dynamics compared to control constructs [52,53]. In recent years, the Biowire system has been further expanded to generate chamber-specific cardiac tissues [41]. It should be noted that not all cardiac maturation studies utilize electrical stimulation, as mechanical force can also be employed to mature 3D heart tissue. Indeed, several recent papers have reported the generation of EHTs between two posts, one rigid, and one flexible, and subjected them to successively increasing stretch in one direction at the flexible end. In general, mechanical stimulation improved the efficiency of excitation-contraction coupling, calcium handling, myofibril alignment, and expression of cardiac maturation markers. However, applying too much stretch led to signs of pathological hypertrophy and fibrosis accumulation [54,55]. Overall, the above studies demonstrate the utility of organoid direct assembly approaches for testing different methods of cardiac maturation. Once matured, these tissues can be used to model adult cardiac pathologies, such as hypertrophic cardiomyopathy (HCM) and heart failure [25,39].

Other groups have generated their own custom platforms to engineer human heart tissue in vitro. In a recent study, Keung et al. [26] generated human ventricular-like cardiac organoid chambers (hvCOCs) to study their responses to various inotropic drugs. First, they grew hESC-CMs as embryoid bodies, resuspended them with human dermal fibroblasts and ECM components, and fabricated them into a custom bioreactor designed to construct hvCOCs resembling a heart pump. In short, the authors took a silicon balloon Foley catheter and filled it with water, then immersed it in an agarose mold. They added the cell-ECM tissue mixture directly to the mold and changed the agarose to culture media after two hours, allowing the hvCOC to spontaneously form around the balloon over the course of 10 days (protocol first developed in [56]). They then exposed these tissues to 25 cardioactive compounds and found that they faithfully recapitulated the expected response to positive inotropic agents, such as isoproterenol (a beta-adrenergic receptor agonist) and levosimendan (a calcium sensitizer) [26]. Finally, one unique study utilized undifferentiated stem cells to generate human chambered cardiac pumps (hChaMPs) in vitro. In this paper, Kupfer et al. [42] took hiPSCs and combined them with an optimized ECM-based bioink that they used to 3D print a structure with two chambers and two flow outlets in two weeks. The presence of the ECM proteins allowed the stem cells to proliferate over 14 days to fill the empty spaces in the construct, after which time they were differentiated to CMs in situ. Remarkably, the 3D structure contained discrete vessels that allowed for perfusion and a continuous muscular wall that formed an enclosed chamber. Longer-term culture allowed the hChaMPs to mature, resulting in functional CMs, calcium transients, action potentials, and beating activity. The authors suggest that these bioengineered heart pumps may be useful for medical device testing and regenerative medicine, demonstrating yet another example of the benefits of using scaffolds to generate 3D human heart tissues [42].

Not all protocols involving directed assembly of organoids use scaffolds to embed their cells of interest. Instead, some utilize hydrogel molds and culture cardiac cells on top, allowing them to form spheres. Varzideh et al. [57] grew organoids from three different cardiac cell types: hESC-derived cardiac progenitor cells (hESC-CPCs), hESC-derived mesenchymal stem cells (hESC-MSCs), and human umbilical vein endothelial cells (HUVECs), cultured at a 10:5:5 ratio. They first coated 24-well plates with Matrigel and allowed it to solidify for 20 min, resulting in a Matrigel bed (first described in [58]). They then suspended the three cardiac cell types in media and cultured them directly on the hydrogels. Exposure to the ECM components present in the Matrigel allowed the suspensions to acquire a three-dimensional rounded shape within 3 days, with maximum beating observed after 7 days. Once the hCOs reached this point, the authors encapsulated them in collagen type I, transferred them into a bio-compatible 3D-printed basket (made of polylactic acid), and sutured the baskets directly into the peritoneal cavity of nude mice. One month after transplantation, the CMs were more mature at the transcriptional and ultrastructural level and the organoids also developed primitive vasculature [57]. In another study utilizing hydrogel molds to direct assembly of 3D structures, Richards et al. [59] made organoids from a 50:50 ratio of CMs to non-CMs (primary human fibroblasts, endothelial cells, and adipose-derived stem cells). Cell suspensions were added to non-adhesive agarose hydrogel molds with 35 microwells each (800 µm diameter) and allowed to settle to the bottom of the mold (protocol first developed in [60]). Microtissue formation and beating were observed within 4 days, at which point the authors subjected them to an oxygen diffusion gradient by placing them in a hypoxia chamber (10% O_2_). They then added norepinephrine to the media to trigger adrenergic stimulation, thereby developing an organoid model of myocardial infarction (MI). The infarcted organoids displayed increased cell death and fibrosis accumulation, abnormal calcium handling, and pathological shifts in their metabolism. Importantly, these hCOs were also used to model chemotherapy-induced cardiotoxicity, potentially providing a platform to recapitulate cardiac responses to various drugs at the tissue level [59].

The studies described above provide ample evidence that directed assembly of human cardiac organoids presents exciting opportunities for disease modeling, drug discovery, and advances relevant to public health. These protocols may even allow us to generate tissues from different germ layers in the same construct, allowing for the study of paracrine interactions between nearby organs. Indeed, a recent study utilized rotary orbital suspension culture of mesendoderm progenitor cells to produce multi-lineage organoids containing cells from the mesoderm (heart) and endoderm (gut) lineages [61]. While directed assembly protocols do have clear applications, it should be noted that many of these studies do not explore the presence of cardiac cell types other than cardiomyocytes and present an oversimplified view of the heart. The human heart is composed primarily of cardiomyocytes (~60%), with the remaining 40% being composed mostly of endothelial cells and cardiac fibroblasts, followed by smaller amounts of smooth muscle cells, epicardial cells, conductance cells, and immune cells [62]. Directed assembly protocols often present few cell types, and cell type ratios that are not close to those observed physiologically. In addition, they all rely on physical structures for proper organization, and despite the fact that some of these materials may recapitulate structural features of the heart, naturally, they are not involved in heart development in vivo. A summary of the recent organoid protocols that rely on directed assembly is presented in Table 1.

## 3. Self-Organizing Human Heart Organoids

Self-assembling organoids are those where cell aggregates undergo differentiation and organization into an organ-like structure with minimal external intervention. They typically require stem cells (either PSCs or adult) with high differentiation potential and addition of specific developmental signals. Good examples of this category of organoids are brain organoids [4], intestinal organoids [16], and kidney organoids [21] to name a few. Heart organoid generation has only been reported from PSCs (either iPSCs or ESCs). Self-assembling heart organoids are expected to form cardiac cell lineages and acquire morphology and function of the heart without directed formation strategies such as cell type co-culture, structural confinement, or scaffold support. It is important to note that some “self-assembling” organoid protocols described here do include the addition of other cell types at a later stage of development, such as epicardial or epicardial progenitor cells in an effort to model developmental processes (such as the generation of the proepicardial organ during heart formation).

Formation of PSC aggregates can be performed by either high-density cell seeding, allowing 3D aggregates to form and transferring them to ultra-low-attachment wells [63], or by seeding the cells into round bottom ultra-low-attachment plates to allow 3D aggregate formation by undisturbed incubation [44,64] or centrifugation of the plate [45,46,47]. For aggregates prepared in 96-well plates, the number of seeded cells ranges from as little as 300–700 cells per well [64] to up to 10,000 cells per well [46]. Most protocols initiate differentiation of the PSC aggregates 2 days after seeding [45,46,63,64], except for Hofbauer et al., which starts after 1 day [47], and Lee et al., which starts after 4 days [44]. These protocols yield a range of cardiac organoids mimicking various aspects of cardiogenesis. While these organoids demonstrate some overlap in the resulting features, there are key differences between them when it comes to cell lineages, morphological structure, functional capabilities, and demonstration of translational applicability. Several of the reported protocols used mouse ESCs (mESCs) [44,63,64], while the most recent ones used human iPSCs or ESCs [45,46,47], with the added benefit of pre-clinical and translational applicability to humans.

Andersen et al. [63] generated precardiac organoids to model the formation of the first and second heart fields using transgenic mouse ESC lines. They tested the effects of various concentrations of the growth factors BMP4 and Activin A on cardiogenesis and found that variations in BMP4 concentration significantly affected organoid formation, while the effect of Activin A variation was only moderate. This finding illustrates the importance of such a platform to facilitate the study of morphogens in cardiac development. Using their transgenic line containing an Hcn4/GFP reporter for the first heart field (FHF) and a Tbx1/RFP reporter for the second heart field (SHF), they were able to demonstrate that FHF progenitors are more likely to give rise to cardiomyocytes (~89%) compared with SHF progenitor cells (~50%). They concluded that the SHF progenitor cells are more likely to give rise to non-myocyte cardiac lineages, which may include cardiac fibroblasts and endothelial cells. They also found that cardiomyocytes arising from the SHF progenitor cells had a delay in differentiation and experienced CM specification up to 2 days after those derived from FHF progenitors, a phenomenon that is also observed in vivo [65]. Lastly, they used their organoids to show that the chemokine receptor CXCR4 can be used to identify SHF progenitors in both mice and humans using an hiPSC cell line. Taking mouse ESC-derived cardiac organoids one step further, Lee et al. [44] cultivated heart organoids with cardiomyocytes, endothelial cells, and smooth muscle cells, with both atrial and ventricular regions. Like Andersen et al., they found that their organoids contain FHF and SHF markers and observed cardiac crescent-like structures in heart development. These organoids, however, develop further than the heart field stage. The authors observed the presence of the atrial-specific cardiomyocyte marker MLC2a and the ventricular-specific marker MLC2v, with a clear spatial separation between them, suggesting atrioventricular specification. Electron microscopy revealed that the ultrastructure was specific to atrial (atria-specific granules, secretory vesicles, and high density of mitochondria), or to ventricular (well-structured sarcomeres, myofibrils containing glycogens) cardiomyocytes, features they also observed in E11.5 mouse heart tissue. Lastly, they analyzed the functional capability of their heart organoids using calcium transient analysis and evaluated electrophysiological parameters using isoproterenol treatment. These studies support the ability of heart organoids to model mouse cardiac development and illustrate their value in understanding novel aspects of morphogen signaling, heart field formation, and atrioventricular specification.

The recent study by Drakhlis et al. [45] used human ESCs to differentiate 3D aggregates embedded in a drop of Matrigel to provide matrix support. The authors found that their organoids form three distinct layers containing lineages beyond those found in the heart. The inner core had endodermal features and was encircled by endocardial-like cells and a dense myocardial layer. The outer layer had endodermal regions and clustered positive for the liver anlagen marker HNF4α. While not exclusively cardiac in nature, these organoids allow the modeling of early heart and foregut development. Aside from cardiomyocyte and endocardial cell markers, they also observed vessel-like endothelial cells, mostly in the inner core. By generating an NKX2.5 knockout (KO) line, they were able to demonstrate that these organoids recapitulate phenotypes of NKX2.5 knockout mice. NKX2.5 is a crucial transcription factor in cardiac development, and its absence is lethal to mouse embryos [66]. Notably, KO organoids appeared less compact, with disorganized sarcomeres and increased cardiomyocyte size. Similar to the organoids produced by Drakhlis et al., the mouse gastruloids produced by Rossi et al. [64] comprise more than just cardiogenic lineages. These gastruloids form a cardiac crescent-like structure adjacent to a primitive gut-like tube. Unlike aggregate generation methods employed by other groups, Rossi et al. seeded under 1000 cells per well in a small volume of media (40 µL) and did not centrifuge the plate, allowing the aggregates to form over 48 h undisturbed, a method reminiscent of an older protocol [67]. Gastruloids display high levels of self-organization and allow for the study of early germ layer interactions. Further development of these gastruloids past the cardiac crescent stage could significantly increase the applicability of this system for the study of diseases affecting cardiac and foregut development in vitro.

Two groups recently reported heart organoids with more complex morphological features than those described above, including chamber formation and self-vascularization [46,47]. Lewis-Israeli et al. [46] generated self-assembling human heart organoids (hHOs) by plating ~10,000 human PSCs per well in 96-well plates, the most cells per aggregate among the reported articles. Using a three-step WNT pathway activation/inhibition/activation, they guided mesodermal induction at days 0–1, cardiogenic mesoderm induction at days 2–4, and a quick, hour-long induction at day 7 to induce proepicardial organ formation. Heart organoids displayed physiologically representative stages of development, showing FHF and SHF markers throughout the protocol as well as the formation of both atrial and ventricular cardiomyocytes (atrioventricular specification). The resulting cardiac organoids had a complex cellular make-up, which included myocardial tissue interspersed with cardiac fibroblasts, epicardial clusters near the outer surface, internal chambers lined with endocardial tissue, and spontaneously forming vasculature. RNA-sequencing revealed that these organoids closely resemble fetal heart tissue and are much more representative of their in vivo counterparts than monolayer models. The authors also evaluated functional electrophysiological features and described several hallmarks of tissue maturation, such as the presence of t-tubules, organized mitochondria, and oxidative metabolism. Lastly, Lewis-Israeli et al. used their organoids to model the effects of pregestational diabetes on the developing heart, showing notable structural, metabolic, and functional differences between organoids cultured in healthy versus diabetic conditions. These results unmistakably illustrate the capacity of these organoids to model human diseases in the fetal heart.

Hofbauer et al. [47] created self-organizing cardioids by seeding 2500–7500 cells in 96-well plates. The method involved a two-step WNT pathway activation/inhibition followed by co-culture of the organoids with epicardial aggregates to simulate the formation of the proepicardial organ. These organoids generated clear chamber-like structures larger in size than those reported by Lewis-Israeli et al. with chamber formation beginning right after mesoderm induction and smaller chambers forming after 2–3 days of differentiation. The authors showed that cardioids can be used as a platform for the study of fetal heart injury by performing cryoinjury, documenting necrosis (using TUNEL staining) and evaluating the contribution of fetal cardiac fibroblasts to the injury.

The recent advancements in self-assembling cardiac organoid technology in the past 5 years greatly increase our capabilities of studying heart development and disease in vitro. These platforms allow stem cell aggregates to differentiate without physical constraints or supporting forces and organize into morphological structures in suspension directed only by the inherent forces of the 3D cellular environment, better resembling in vivo development. By guiding the stem cells onto a mesoderm/cardiac mesoderm path, these organoids give rise to a more diverse population of cells than co-culture protocols allow and facilitate more physiologically relevant cellular interactions. While self-assembling organoids do show a great deal of promise in modeling heart development and disease, they are not without their limitations. The self-assembling nature of the methods to create these organoids allows researchers to guide cell type specifications but does not allow for precise control over the cell-to-cell ratios, the overall composition, or the resulting structural organization. Furthermore, the lack of supporting material results in 3D constructs that are more delicate in nature compared to many of their directly assembled counterparts. Lastly, the cells and tissues present in these organoids are immature and fetal in nature, and therefore, future studies into the maturation of self-assembled heart organoids will be necessary for modeling the adult human heart to explore associated diseases and toxicity studies. A summary of the recent organoid protocols that rely on self-assembly is presented in Table 2.

## 4. Conclusions

The development of the heart is a complex process that utilizes the coordinated effort of a multitude of progenitor cells to give rise to a complex organ. Modeling such a process in vitro is a difficult feat, but in recent years, researchers have made significant advances on this front. Both direct and self-assembled cardiac organoids provide us with indispensable tools to study the process of human heart development and disease in extraordinary detail. Organoids created by direct assembly grant researchers extensive control over the composition and organization of the 3D tissue, allowing the exploration of specific functions and attributing the effects to their cells of interest. Studies using direct assembly have shown progress in cardiomyocyte tissue maturation strategies by employing techniques such as mechanical and electrical stimulation. In contrast to direct assembly strategies, which have been around for well over a decade, the field of self-assembling cardiac organoids is still very much in its infancy but shows enormous promise. Much work remains to be performed, such as further refinement of developmental models and maturation of these organoids beyond the fetal stage to model adult conditions. Both approaches to create heart organoids come with advantages and disadvantages (summarized in Table 3) and both are likely to remain the focus of future studies for some time. Directly assembled organoids are likely to lead the research in heart tissue constructs capable of withstanding high mechanical forces and have a head start on targeting more adult heart tissues and are therefore more ideal for surgical intervention in the form of patches and replacement tissue. On the other hand, self-assembled organoids will likely take the lead in the study of developmental diseases and disorders, and are more capable of physiologically representing the complexity of the human heart as a whole.

Overall, heart organoids open the window into a wide range of basic studies and translational applications. Directly assembled organoids have been used in functional screening, cryoinjury studies, response to COVID-19 infection, and modeling chemotherapy-induced cardiotoxicity and myocardial infarction. Self-assembled organoids have been used to investigate heart field progenitor cell markers, the effects of gene knockouts (e.g., *NKX2-5*), the response to isoproterenol compared to that observed in vivo, cryoinjury studies, and disease modeling (pregestational diabetes). Future steps will aim to improve developmental modeling by refining protocols and lengthening the viability of the organoids over time, as well as creating more mature heart organoids to closely resemble the adult heart, thus opening the field to a range of applications beyond fetal heart development and disease.

## Figures and Tables

**Figure 1 biomolecules-11-01277-f001:**
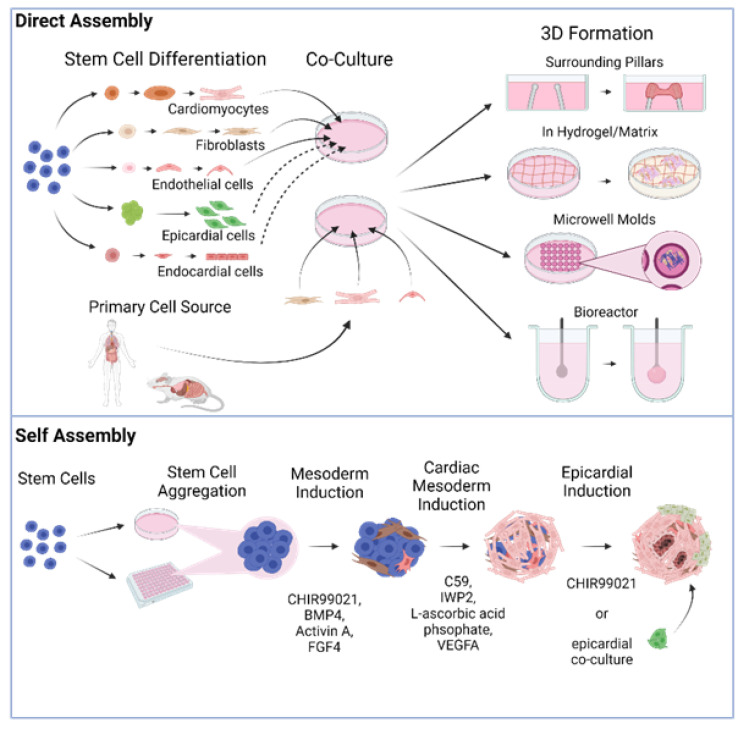
Different fabrication approaches for cardiac organoid generation in a dish.

**Table 1 biomolecules-11-01277-t001:** Cardiac Organoids by Direct Assembly: techniques, characteristics, and applications.

Reference	Cell Source	Aggregation Technique	Functional Assessment	Translational Applications
Mills et al., 2017	hiPSC-CMs, Stromal cells	Heart-Dyno: Grow hCOs around two elastomeric posts	Force/contractile analysis, electrophysiology, calcium imaging, metabolic profiling	Cardiac maturation studies
Mills et al., 2019	hiPSC-CMs, Stromal cells	Heart-Dyno: Grow hCOs around two elastomeric posts	Force/contractile analysis	Drug screening: Regenerative responses
Mills et al., 2021	hiPSC-CMs, Stromal cells	Heart-Dyno: Grow hCOs around two elastomeric posts	Force/contractile analysis	Modeling cardiac effects of COVID-19
Voges et al., 2017	hESC-CMs, Stromal cells	Grow hCOs around two elastic exercise poles	Force/contractile analysis	Cardiac regeneration studies
Ronaldson-Bouchard et al., 2018	hiPSC-CMs, Human DFs	Grow human cardiac tissues around flexible elastomeric pillars	Force/contractile analysis, electrophysiology, calcium imaging, metabolic profiling	Cardiac maturation/adult disease modeling
Keung et al., 2019	hESC-CMs, Human DFs	Grow hvCOC around silicon balloon Foley catheter (custom bioreactor)	Force/contractile analysis	Drug screening: Inotropic responses
Kupfer et al., 2020	hiPSCs-CMs	3D print hChaMP, culture hiPSCs on construct, differentiate in situ	Force/contractile analysis, pressure/volume analysis, calcium imaging, optical mapping	Medical device testing/tissue grafting
Varzideh et al., 2019	hESC-CPCs, hESC-MSCs, HUVECs	Culture on Matrigel-coated hydrogel	Electrophysiology, voltage-sensitive dye imaging (VSDI)	Transplantation into mice
Richards et al., 2020	Human CFs, HUVECs, hADSCs	Culture in microwells containing agarose hydrogels	Calcium imaging, metabolic profiling, mechanical testing	Myocardial infarction modeling
Devarasetty et al., 2017	hiPSC-CMs, Human CFs	Culture in round-bottom well plates, immobilize in hydrogel	Force/contractile analysis	Drug screening: Chronotropic responses
Buono et al., 2020	hiPSC-CMs, HCMECs, Human CFs	Invert cell suspension, gravity-enforced aggregation (“Hanging Drop”)	Calcium imaging	Modeling hypertrophic cardiomyopathy
Silva et al., 2020	hiPSC-mesendoderm progenitor cells	Rotary orbital suspension culture	Electrophysiology, calcium imaging	Modeling multi-tissue interactions

**Table 2 biomolecules-11-01277-t002:** Cardiac Organoids by Self-Assembly: techniques, characteristics, and applications.

Reference	Cell Source	Aggregation Technique	Differentiation Protocol	Cell Types Observed	Stage of Development	Translational Applications
Andersen et al., 2018	mESCs, hiPSCs	high-density cell seeding	D0: BMP4/ActA (40 h)	Cardiac progenitor cells, FHF cells, SHF cells, cardiomyocytes, endothelial cells, smooth muscle cells, fibroblasts	Precardiac heart field specification. No beating reported.	Discovery of CXCR4 as SHF progenitor marker in human organoids
Lee et al., 2020	mESCs	round-bottom, low-attachment well plate	D0-D9: FGF4D9-D15: FGF4, BMP4, BIO, LIF	Cardiac progenitor cells, FHF cells, SHF cells, cardiomyocytes (atrial), cardiomyocytes (ventricular), smooth muscle cells, endothelial cells	Atrioventricular chamber specification. Representing embryonic day E7.5-E9.5. Beating after 10 days.	
Drakhlis et al., 2021	hESCs	round-bottom, low-attachment well plate with centrifugation, embedded in Matrigel	D0: CHIR99021 (24 h)D3: IWP2 (48 h)	Cardiac progenitor cells, cardiomyocytes, mesenchymal cells, endothelial cells, endocardial cells, liver anlagen	Cardiac mesoderm and foregut endoderm specification. Beating after 7–10 days.	NKX2.5 knockout recapitulates in vivo congenital heart defects
Rossi et al., 2021	mESCs	round-bottom, low-attachment well plate	D0: CHIR99021 (72 h)D4: bFGF, VEGF165, L-ascorbic acid phosphate (48 h)	Cardiac progenitor cells, FHF cells, SHF cells, cardiomyocytes, endothelial cells, endodermal cells	Cardiac crescent and gut-like tube. Beating after 6 days.	
Lewis-Israeli et al., 2021	hiPSCs,hESCs	round-bottom, low-attachment well plate with centrifugation	D0: CHIR99021/BMP4/ActA (24 h)D2: Wnt C-59 (48 h)D7: CHIR99021 (1 h)	Cardiac progenitor cells, FHF cells, SHF cells, cardiomyocytes (atrial), cardiomyocytes (ventricular), epicardial cells, endocardial cells, endothelial cells, cardiac fibroblasts	Heart field and atrioventricular specification and chamber formation. Fetal-like heart organoids transcriptomically similar to human fetal gestation day 57–67. Beating after 6 days.	Modeling pregestational diabetes-induced congenital heart disease
Hofbauer et al., 2021	hiPSCs,hESCs	round-bottom, low-attachment well plate with centrifugation	D0: CHIR99021, ROCKi (36–40 h)D2: VEGF-A (96 h, with medium change every 48 h)D8.5: co-culture with epicardial aggregates	Cardiac progenitor cells, FHF cells, cardiomyocytes (atrial), cardiomyocytes (ventricular), endothelial cells, epicardial cells, endocardial cells, fibroblasts	First heart field specification and chamber formation. Beating after 7 days.	Cardiac injury model

**Table 3 biomolecules-11-01277-t003:** Direct Assembly vs. Self-Assembly: advantages and disadvantages.

Direct Assembly		Self-Assembly	
Advantages	Disadvantages	Advantages	Disadvantages
Precise control over cell composition Relative control over size and shape Can comprise of more mature cells Pillars can be used for mechanical stimulation Scaffolding allows for stronger tissue construct	Often only include cardiomyocytes or a small number of cell types Rely on physical structures for support Not as physiologically relevant to heart development and structure Miss out on developmental stages of the early heart	Embryoid body differentiation more closely represents physiological heart development Multi-cellular composition High tissue complexity Can be used to model various stages of early heart development Self-organizing capabilities unhindered by physical restraints	Minimal control over cell type composition Minimal control over shape Comprised of immature (fetal-like) cells Constructs are more feeble in nature

## Data Availability

Not applicable.

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
