# Peer review of "Heart Organoids and Engineered Heart Tissues: Novel Tools for Modeling Human Cardiac Biology and Disease"

_biomolecules, 2021, doi:10.3390/biom11091277_

Round 1
Reviewer 1 Report
Lewis-Israeli et al discuss in this review methodologies for engineering cardiac tissues and generating heart organoids. While this review paper could potentially be valuable for the community it would benefit from having a more compelling discussion in each section. In addition, the review is missing some key studies in the field of cardiac tissue engineering and I would expect the authors to add more references to make it into a more comprehensive review.
Minor Comments:
- Page 2 line 59: I suggest the authors would replace organ-like structure with tissue like structure and functionality.
- Page 2 line 64-66: I suggest the authors would rephrase this sentence. It currently is misleading and reads as if the pillars and bioreactors are considered scaffolds as well.
- Page 2 line 68: Early studies in cardiac tissue engineering in the late. 1990s to early 2000s did not use human heart cells.
- Page 2 line 70. A lot of the early work in cardiac tissue engineering did not necessarily engineer only cardiac spheroids. I would suggest the authors to discuss this. In addition certain pioneering studies in the field are missing in this manuscript. For example:
- Bursac, N., et al. (1999). Cardiac muscle tissue engineering: toward an in vitro model for electrophysiological studies. Am. J. Physiol.
- Boudou, T., et al (2012). A microfabricated platform to measure and manipulate the mechanics of engineered cardiac microtissues. Tissue Eng. Part A.
- Tiburcy, et al. (2017). Defined Engineered Human Myocardium With Advanced Maturation for Applications in Heart Failure Modeling and Repair. Circulation 135, 1832–1847.
- Zhao, Y., et al. (2019). A Platform for Generation of Chamber-Specific Cardiac Tissues and Disease Modeling. Cell.
- Page 5, line 217: I would suggest the authors to rephrase this sentence as scaffolds and biomaterials are important to support cardiac tissue assembly (Fleischer et al. Current Opinion in Biotechnology (2013) and Biomaterials (2013)).
Major comments:
- Page 3, section 2: The title of this section is misleading and in general the terminology “directed assembly of human heart organoids”. Engineered tissues and organoids are inherently different therefore I would suggest to make these definitions as distinct as possible throughout the manuscript. In addition, I would suggest the authors to discuss one of the burning issues in the field; cardiac tissue maturation and how approaches in cardiac tissues engineering address it. While the paper by Ronaldson-Bouchard is discussed, the maturation aspect is not mentioned. I would suggest to further discuss maturation methodologies such as mechanical stimulation and electrical stimulation and their advantages.
- A table comparing advantages and disadvantages of each methodology would be helpful.
- Section 3 should have a more comprehensive discussion about the limitations of organoids.
Reviewer 2 Report
This manuscript was reviewed the heart organoids by summarizing with assembly methods.
Direct assembly was reviewed that tissue engineered by with scaffold or culture materials, culture technology. Self-assembly was indicated differentiation of embryonic heart development. It was interesting and well referenced many reports. It may be important point in direct assembly groups how to assess the cardiac function. It was only recommendation, but it will be more understandable table to add what can be assessed with the models in Table1.
Second, self-assembly also indicated interesting groups, it may be important the stage of heart development. It was also simple recommendation to add the stage of development (refer the embryonic date, beating or not) in Table 2.
And in conclusion section, it will be better to perspective future about heart organoid by authors. ( Which will be progressed, Direct or Self?)
